# Experimental Study of the Surface Quality of Form-Cutting Tools Manufactured via Wire Electrical Discharge Machining Using Different Process Parameters

**DOI:** 10.3390/mi14111976

**Published:** 2023-10-24

**Authors:** Amir Alinaghizadeh, Mohammadjafar Hadad, Bahman Azarhoushang

**Affiliations:** 1Department of Mechanical Engineering—Manufacturing Group, Kish International Campus, University of Tehran, Kish 1591634311, Iran; 2Institute of Precision Machining (KSF), Hochschule Furtwangen University, 78532 Tuttlingen, Germany; amir.alinaghizadeh.abiazani@hfu.eu; 3School of Mechanical Engineering, College of Engineering, University of Tehran, Tehran 1417466191, Iran; mjhadad@ut.ac.ir; 4Department of Mechanical Engineering, College of Engineering and Technology, University of Doha for Science and Technology, Doha P.O. Box 24449, Qatar

**Keywords:** form machining, wire electrical discharge machining, form-cutting tools, recast layer, surface quality

## Abstract

Form-cutting tools are an economical choice for turning parts with defined profiles in mass production. The effects of the form contour of these tools—produced by the wire electrical discharge machining (WEDM) process—on tool quality were investigated in this research. This study focuses on reducing the adverse effects of the recast layer induced by WEDM on form-cutting tools. The basic component types of profile forms in form-cutting tools can be summarized by a combination of four modes, i.e., concave and convex arcs and flat and oblique surfaces. Hence, sample cutting tools with three different radii of convex and concave arcs and a flat surface were produced. During the WEDM operation, one-pass mode was used for roughing, two passes for semi-finishing, and three passes for finishing. Furthermore, the difference between the percentage of oxygen and carbon elements on the recast layer in the two areas above the workpiece or wire entry point and the bottom area of the workpiece or wire exit point was investigated. Finally, the influences of the direction, size of the curvature, and the number of passes in the wire electric discharge process on the recast layer were analyzed. It was observed that the recast layer thickness could be reduced by increasing the number of WEDM process. Additionally, the uniformity of the cutting contour was superior in the entry region of the wire going into the workpiece compared with the exit region of the wire.

## 1. Introduction

The first topic that must be addressed as an introduction is enhancing precision and productivity through the utilization of form-cutting tools in profile-turning operations. Form-cutting tools are used to generate specific forms on a workpiece in the profile-turning process. They are essential in a wide range of manufacturing industries. They are required for machining complex geometries and as special tools for mass production. Substituting various standard tools with a form-cutting tool leads to quality improvements, time saving, cost efficiency, etc. [1,2]. One form-cutting tool can handle many diameters and avoids the need to index the turret. The geometry, dimensional accuracy, and the quality of the cutting surface of form-cutting tools have a direct effect on the precision, productivity, and quality of machined workpieces [3]. Although machining forces are high because of the large contact area that the form-cutting tool has with the workpiece, this method is a suitable alternative to the point-to-point turning process using machine-controller interpolation. Form-cutting tools have a large contact area with the workpiece and may cause poor workpiece quality because of vibrations and chatter formation [4]. Low cutting speeds, high feed rates, and being under workpiece pressure during clamping are among the reasons for these problems. Additionally, chip formation is not uniform in the profile-turning process, and in some areas, according to the shape of the workpiece profile, the chips are aggregated or separated and far away from each other [5]. Another disadvantage of using form-cutting tools is that the feed into the workpiece is usually slow because of the width of the tool. However, form-cutting tools can generally reduce production costs and increase productivity as well as process accuracy and reliability.

The second topic that must be addressed as an introduction is optimizing machining performance through insights into the characteristics and applications of form tools in the cutting processes. Cutting with form tools is defined as a process in which two or more cutting edges perform a cutting action simultaneously with mutual interactions between their respective force systems [6,7]. These tools have one or several cutting edges with a precisely defined negative profile or contour in the workpiece surface. Straight-form tools are useful for machining deep, rectangular-form grooves. However, given the mixed chip flow, cutting is severely limited. To achieve a satisfactory surface finish, this type of form tool must be used at rather slow cutting speeds. Wide-form tools generate more heat and are generally prone to chatter [8]. The cutting force is thought to be unrelated to the depth of the cut but proportional to the engagement length of the cutting edge [6,7].

The third aspect worthy of consideration in the introduction of the current paper is the ample justification for the strategic selection of high-speed steel as a pragmatic choice in modern cutting tools in the field of mechanical engineering. Nowadays, cutting tools are mostly made of cemented carbides. But still, there are logical reasons for using high-speed steel (HSS) as a cutting-tool material, some of which are as follows:▪Reasonable behavior and performance at low cutting temperatures, which could also be applied to form-cutting tools given their low rotational and cutting speed.▪Low prices compared with carbides (cost-effective for prototyping applications).▪Fewer manufacturing challenges because of their lower hardness (compared with other cutting materials).▪Sufficiency for primary machining tests used to inspect the geometric accuracy of the form contours produced (first on the tool and then on the workpiece).▪Suitability for machining non-ferrous metals.▪Excellent toughness compared with other cutting materials.▪Adequacy for manual machines (depending on the influence of the operator’s skills on the quality of the process and the possibility of human error).

The fourth section highlighted in the introduction of this paper pertains to a comparative analysis of previous research in the field, focusing on mitigating recast layer effects through an in-depth exploration of WEDM process parameters on the surface quality and form accuracy of HSS form-cutting tools. A manufacturing method for producing form-cutting tools suitable for single production and prototyping involves using electrical discharge on a wire-cut machine (WEDM), allowing the required features to be created on the tool with a rather low production time and high accuracy [9,10]. This method can be combined with CAD/CAM programs. Usually, the electric discharge method is used to produce form inserts of between 5 and 20 pieces. This number is determined based on the complexity of the initial machining tests or the accuracy of the initial settings. However, machined surfaces with poor surface integrity are the main limitation of the WEDM method [11]. The WEDM process can be divided into various passes, from roughing to fine finishing. In the case of rough machining, more thermal energy is transferred to the workpiece material. Hence, the re-solidification of the material or a recast layer can be observed on the cut area [12,13]. Therefore, the machined surface shows high surface roughness values. However, in the case of fine cutting (also known as trimming [14]), very fine electrical discharges are produced that melt only a tiny amount of material on the machined surface, which is easily removed by pressure waves [15,16,17]. Various researchers studied the effects of WEDM parameters on the quality of parts made of HSS. In one investigation, it was attempted to subject HSS plates of various thicknesses to EDM to determine the optimum value of current as a function of plate thickness. Experiments were undertaken at various levels of discharge current, and the resulting spark gap, material removal rate, and surface roughness were measured. Additionally, the interaction of process parameters on the surface roughness of various plates was investigated. Artificial neural network (ANN) and support vector machine (SVM) models were established and programmed with experimental data to forecast the best process parameters [18]. The main objective of a separate research work was to study the effects of different WEDM parameters on the machining quality and to obtain the ideal sets of process parameters such as pulse-off time, pulse-on time, taper angle, part thickness, wire tension, servo voltage gap, and wire feed while taper cutting on HSS. It was found that an increment in part thickness, taper angle, and wire feed leads to a continuous increase in surface roughness. Reduced pulse-off time resulted in a continuous decrease in surface roughness [19]. Chandrakanth et al. also studied the WEDM of M42 HSS grade; they found that the main factors affecting the metal removal rate were pulse-on time spark voltage, and the combination of these two parameters also played a significant role. In their experiments, the wire tension has a considerably smaller role in the metal removal rate of the workpiece [20]. Another study evaluated the dimensional deviation of parts made of M42 HSS and machined in a WEDM process with cryogenically treated brass wire. The ANOVA analysis of the dimensional deviation showed that pulse-on time, pulse-off time, and spark gap voltage were the most significant variables. Furthermore, the following conclusions were achieved: pulse-on time (Ton) and pulse-off time (Toff) showed more significant effects on the dimensional deviation of the parts. The shallow cryogenic treatment effect on wire electrodes in WEDM led to a pronounced enhancement of the dimensional accuracy of the parts since the superficial cryogenic treatment improved the electrical conductivity of the brass wire electrode [21]. Puri et al. examined the wire lag occurrence in WEDM and the tendency of change in the geometrical accuracy due to the wire lag with various machining parameters. The optimal parametric adjustments for several machining conditions showed that the essential contributors to the geometrical accuracy due to the wire lag were pulse peak current, pulse-on time, and pulse-off time in the roughing stage and wire tool offset, wire tension, pulse peak voltage, servo spark gap adjusted voltage, constant cutting speed, and wire tool offset in finishing stage [22]. Farooq et al. machined convex and concave profiles in Ti6Al4V using the WEDM process. The effects of wire feed, servo voltage, pulse-off time, and pulse-on time on the geometric accuracy of convex and concave profiles and corner radiuses were extensively studied. They found that an optimized combination of machining parameters is required to achieve the lowest geometric deviations (overcutting for convex profiles and undercutting for concave profiles) and corner radiuses. In conjunction with the optimized parameters, the provision of wire offsets in the range of 0.169 to 0.173 mm could further reduce the geometric deviations of the actual machined profiles from the designed geometries [23]. Chen et al. aimed to explore and minimize the geometric accuracy of rough corner cutting. The main factors affecting corner accuracy (135°, 90°, and 45° angles) were investigated in depth, and an elliptical fitting technique was developed to explain the trajectory of the center of the wire electrode, and the applicability of the model was verified by measuring the corner edge of the component. Their work revealed that the wire deflection and vibration were the primary reasons for right-angle and obtuse-angle corner cutting, and the discharge concentration phenomena also had an essential impact on acute-angle corner cutting [24]. Pramanik et al. examined the geometric errors like circularity, cylindricity, and diametrical errors of a hole machined with the WEDM process of Ti6Al4V alloy via varied flushing pressure, tension in wire, and pulse-on time. It was observed that the circularity error of the holes decreased with the increase in the flushing pressure and the tension in the wire, whereas it decreased with the increase in the pulse-on time—the cylindricity error diminished with the rise in pulse-on time, flushing pressure, and wire tension. Lastly, the absolute diametral error decreased with the increase in wire tension and pulse-on time, although the absolute diametral error increased with the rise in pulse-on time [25]. In another study, a survey of the sources of dimensional deviations in small-radius circles was conducted by Conde et al. The research concentrated on two influencing factors: the wire lag (deflection) and the geometry of the concavity on the component. Based on the results, a categorization of the radius was suggested, taking into account the main reason for the poor accuracy. It was concluded that the wire lag requires consideration only when machining circles with radii less than a specific value [26]. This is significant since one of the largest challenges in the WEDM process is the deviation in cutting small-radius corners. In the study by Firouzabadi et al. [27], the research concentrated on small-radius concave and convex corner errors due to the poor rigidity of the wire and processing forces affecting the wire. They proposed alternative solutions in the event of consecutive cuts in the roughing and finishing stages. Preliminary tests were performed using the frequency of discharges and the feed rate as input parameters and the residual material on the cutting paths as the output variable. The acquired results showed that roughing is the most significant aspect of the WEDM cutting process, and optimizing the input variables had a more significant effect on the control of the thickness of the residual material on the flat paths than on the convex corners with little radius [27]. 

As a research gap or problem statement, it must be stated that the influence of the WEDM process parameters on the surface quality of parts made of HSS and their form accuracies, considering various contours, has yet to be studied. The present study is focused on reducing the adverse effects of the recast layer induced by WEDM on the form-cutting tools made of HSS. The primary component types of profile forms in form-cutting tools are summarized by combining four modes, i.e., concave and convex arcs and flat and oblique surfaces. Form-cutting tools made of HSS were fabricated with three different radii of convex curvature, three different radii of concave curvature, and a flat surface. During the WEDM operation, one-pass mode was used for roughing, two passes for semi-finishing, and three passes for finishing. Furthermore, the surface quality of the recast layer at two areas, namely the wire entry point and the wire exit point, was studied. Finally, the effect of the direction, size of the curvature, and the number of passes in the wire electric discharge process on the recast layer on the surface of the form-cutting tools were analyzed.

## 2. Materials and Methods

The form-cutting tools for turning machines are categorized into three main groups: convex, concave, and flat surfaces. Figure 1 shows a sample with different profiles provided by a single form-cutting tool. This figure shows that each form profile can be provided and generalized if the turning process is performed with a suitable cutting surface.

High-strength steel (HSS) was used as the material of the form-cutting tools. The chemical composition and material properties of utilized HSS are shown in Table 1.

A bulk HSS was chosen as the base material for this study, and all possible form contours were fabricated on the base material. Figure 2 shows the utilized WEDM machine and the produced form-cutting tools. The classification system delineates various surface curvatures: A designates concave curvature with a big radius, B represents concave curvature with a moderate radius, C signifies concave curvature with a small radius, E denotes a flat surface, G indicates convex curvature with a big radius, H corresponds to convex curvature with a moderate radius, and I is assigned to convex curvature with a small radius. Parts D and F are the result of additional cuts that were created to eliminate the convex and concave states caused by the geometry of parts G and C, and they do not play a role in the design of the experiments conducted for this research work. The experiments were carried out on a universal wire-cut electrical discharge machining tool (Mitsubishi MX600, German Branch, Mitsubishi-Electric-Platz 1, D-40882 Ratingen, Germany).

To examine the effectiveness of WEDM passes, three different strategies, i.e., single pass for the rough machining, double pass for the semi-finishing process, and triple pass for the finishing process, were defined. Figure 3 shows the WEDM machine path in these three different strategies, and Table 2 shows the utilized parameters of the WEDM process.

A scanning electron microscope (SEM) was used to investigate the effect of the recast layer. The cut samples were observed from different angles, and the results were compared regarding the surface morphology and kurf analysis. Figure 4 depicts the selected directions for image scanning and the utilized SEM instrumentation.

A confocal microscope was used to examine the surface characteristics of the recast layer (Figure 5). A sine vise was used to obtain a suitable photography angle concerning the recast layer surface. The direction was adjusted so that there was parallelism between the direction of photography and the normal vector of the recast layer surface.

Figure 5 shows the three-dimensional plan of different samples and the specified location for photography. Due to the limitation in number, the middle area of the samples was chosen for microscopy so that the distances to the entry and exit areas of the wire are equal. A Hommel Tester stylus profilometer was used throughout the experimental studies to determine each sample’s geometrical accuracy (surface profile) (Figure 6).

## 3. Results and Discussion

### 3.1. Influences of Number of Passes and Different Contour Profiles on Surface Quality

The SEM observation revealed that using single-pass machining, a high density of melted debris, dimples, and microvoids was induced on the machined surface. Figure 7 shows the morphologies of the machined surfaces induced by different WEDM strategies. A recast layer is commonly observed due to the melting and re-solidification of material. It can be seen that the mentioned surface characteristics have been significantly reduced in the cut by two passes (semi-finish). The recast layer formed on the machined patterns for modes with one, two, and three passes indicates significant thermal damage in the case of rough-cut mode. That is because, using the rough cut method, the discharge pulse has a high pulse-on time, increasing the spark intensity, melting more material, and forming larger voids on the machined surface. A part of molten material is flushed away by pressurized dielectric fluid; however, some air bubbles become entrapped in it. When the remaining molten material re-solidifies, these air bubbles collapse and generate microvoids on the machined surface.

Also, regarding the cut with three passes (finishing), regardless of the geometry of the sample, which has a flat surface or concave and convex curves, the ultrafine discharge pulses melt only a tiny amount of material, which can be easily flushed away by pressurized dielectric fluid generated in the absence of electro-discharge channel. Hence, recast layer formation is significantly reduced in finishing cut mode. On the other hand, the propensity of micro-globule and crater formation is considerably reduced in finishing cut mode due to ultrafine electrical discharge except for a few micro-holes and melted debris. Moreover, Figure 7 shows the micro/nanocavities formed on the machined surface.

Another critical point is the effect of the length of the sector of the wire engagement with convex, flat, and concave surfaces, increasing from low to high. During the more extended engagement sector, each point of the surface is exposed to electrical discharge for a longer time, and its temperature increases more. Therefore, it is more susceptible to thermal damage while being cooled by the dielectric fluid. In addition, the melted debris and deposits are distributed on the surface. A more extended period of exposure to heat causes spherical patterns of particles and frozen deposits on the surface that are mentioned in other articles with spherical and irregular debris names. Figure 8 shows significant differences among flat, convex, and concave samples. These differences in surface characteristics can be seen for all electrical discharge conditions of the samples.

### 3.2. Surface Characterization of the Recast Layer

#### 3.2.1. Comparative Analysis of Geometric Variations in CLSM 3D Surface Spectrum

Figure 9 shows a view of the examined sections of seven samples with different geometries regarding the contour of the form, which was created in three-dimensional form using the color spectrum. This image was created in connection with samples that were made through three passes of cutting with wire cut wire. The square sections that have been photographed have an area of 2.56 mm^2^ (1.6 mm × 1.6 mm), and to prevent any measurement error resulting from graphic matching or matching movements in the longitudinal and transverse axes of the microscope, leveling mode was avoided. As mentioned earlier, the middle area of the samples was used for photography to ignore the differences in their entrance and exit areas. Figure 10 compares the samples obtained from one, two, and three cutting passes in the electrical discharge process. Geometrically, these samples have a concavity with an average radius among the radii of the projected curve. As the color spectrum shows, in the sample made with one pass, due to higher energy density (in the electrical discharge process), the height of the peaks and valleys are more significant on the surface of the recast layer. The direction of wire movement in these samples is parallel to the axis of the curvature, and it is visible in the picture, but no unique direction can be seen in the voids on the surface. Therefore, in the construction of cutting tools, the selection of the wire path can only be selected based on the geometrical limits of the form contour.

The peaks in the sample made with a single cutting pass through the electrical discharge process, in Figure 10, clearly show that the higher wear observed in the tools made by a similar method is a result of their removal due to mechanical contact. It can be said that these peaks, in the order of proximity to the cutting edge due to continuous contact with the workpiece, gradually decrease in height one after the other, and by removing them, other peaks become involved with the surface of the workpiece as new friction factors. This process continues until it is no longer possible to use it according to the definitions given for the tool’s lifespan.

#### 3.2.2. Analysis of Wire Electrical Discharge Machining (WEDM) Effects on Surface Properties and Geometric Deviations in Convex and Concave Curves

The length of the wire engagement (contact) segment with convex, flat, and concave surfaces increases from low to high. This case is shown in Figure 11 and Figure 12 for convex and concave samples with different radii of curvature. For a more extended engagement (contact) sector, each point on the surface is subjected (exposed) to an electrical discharge for a longer (more extended) time, and its temperature increases more. Therefore, it is more susceptible to thermal damage when cooled by the dielectric fluid. In addition, prolonged exposure to heat leads to spheroidization of the molten material at the cutting region (zone), spreading frozen particles and deposits on the surface. From this point of view, there is a significant difference between the SEM images of the samples.

In all the SEM images, the degree of sphericity of the visible particles on the surface decreases from sample C to sample A, and their size decreases in concave samples compared to convex samples. In convex samples, two divergent convex surfaces are in contact with each other during the electrical discharge, one related to the wire and the other to the samples. Therefore, there are fewer areas of engagement (contact) than with concave samples. This means they are exposed to heat for less time, making all surface constituents finer-grained. These differences in surface properties can be observed for all electrical discharge conditions of the samples.

The necessary measurements were made with a profile meter to investigate the effects of the number of passes in wire EDM on geometric accuracy concerning the form contours. As shown in Figure 13, the results showed that the highest deviation from the form profile for both convex and concave arcs (curvatures) occurred for cuts with one pass, and the lowest deviation from the profile occurred for cuts with three passes. Another result is that the profile created for concave arcs has a deviation compared to the amount of nominal concavity. According to Figure 13, the actual profile has more depth or less transverse extension than the nominal profile. In other words, it results in the inverted shape of a dome. In the case of convex curves, conversely, there is a deviation compared to the numerical value of nominal convexity. Also, in this case, the accurate (actual) profile has a lower height than the nominal profile. In other words, it was created in the form of a dome. The number of deviations is inversely related to the number of passes. The deviation trend is influenced by the sector length of the contact between the wire and the workpiece; the type of the curvature, i.e., convexity or concavity; and the amount of the radius of convexity or concavity, which is decisive, as explained earlier.

Figure 13 shows an exaggerated representation of the cutting with one, two, and three passes for a better understanding of the deviation from the nominal form profiles for concave and convex curves. Also given is the amount of the geometric deviations for B and H samples, which were related to the middle size of the radius for concave and convex curves, respectively. At the points where the most significant deviation had occurred, the actual profile between the circle corresponding to the nominal size of the arc and the circle generated by the drawing method with three reference points is shown in corresponding magnification. The green arcs represent the nominal profile, and the pink arcs represent the actual profile. For both concave and convex curves, the difference between the geometric deviation was much more extensive for cuts in one and two passes than for the difference between the geometric deviation cuts in two and three passes. In other words, the second cut pass plays a more critical role in the geometric correction of the form profile than the third pass.

Since the radius’s magnitude of each sample’s curvature was considered to be different, the amount of geometric deviation of each sample depended on the radius of that sample. To eliminate this dependence and obtain an independent value for the geometric deviation, the deviation value for each sample was divided by the sample’s radius, and the geometric deviation’s magnitude was calculated as a percentage in the comparisons. To obtain the geometric deviation value for each sample, the profile of the WEDM cutting surface was first measured using a profile meter device and extracted in “DXF” format. Then, via the three-point drawing method, a circle was applied (fitted) to each profile. The difference between the circle radius fitted to the actual profile and the nominal radius for each sample was considered the radial geometric deviation value.

Figure 14 shows the comparison of the radial geometric deviations for different samples. Since the radius of curvature of the surface of the flat sample was infinite, it was not considered in the diagram. The figure shows that the magnitude of the radial geometric deviation is reduced for samples cut with three passes compared to samples that were cut with one pass. The sign of the geometric deviations of the convex samples was negative, and the sign of the geometric deviations of the concave Flat samples was positive. This means that for convex samples, the radial geometric deviation caused a decrease in convexity and for concave samples caused an increase in concavity. The reason for this behavior has already been explained. The magnitude of the deviations was greater for convex samples than concave samples. As the radius of curvature increased in both convexity and concavity states, and the surface became more similar to the flat surface, the amount of deviation in the geometric profile of the surface decreased. Although the values of the deviations are not the same for the direction of convex and concave curvature, the trend of almost equality can be seen in the behavior of the geometric deviation of the surface profile of the samples, which are the opposite of each other in terms of being positive and negative. There are differences in the geometric deviation of the surface profile of samples cut with a different number of passes. In the case of samples cut with one and two passes, the mentioned difference is much more than two and three passes. Therefore, Figure 14 also confirms that the second pass significantly affected the geometric correction of the surface profile compared to the third pass. Although the mentioned figure was obtained for HSS material, there are certain limitations in using it for other workpiece materials due to the different behaviors they are likely to have. It is possible to calculate the offset used in the different numbers of passes of the WEDM, depending on the direction and dimension of the radius in the curvature of the cutting tool contour. By interpolation, it is also possible to obtain the value of the surface profile’s geometric deviation for a specific curvature radius.

#### 3.2.3. Surface Roughness Analysis of Wire Electrical Discharge Machined Samples with Varied Geometries and Passes

Results for the samples made with seven different geometries for the form profile and three electrical discharge methods in terms of the number of cutting passes are shown in Figure 15. It can be seen that the roughness values of the samples produced with three-pass cutting are reduced compared to those with one pass. The descending trend indicates the effect of energy density and adjustment parameters between the first and third passes on the height of the recast layer surface. The roughness distribution in the convex curvature samples was lower than in the concave ones. The flat sample has the worst surface roughness in all three machining methods. In convex samples, the surface roughness value increases with the decrease in the radius of convexity. This process is the same for a sample with concave curvature. The difference between the roughness of the samples with a medium and large radius of curvature is much smaller than the difference between the roughness of the samples with a medium and small radius of curvature.

The recast layer can be very effective in generating surface roughness. The result related to the roughness of the different samples showed that the flat sample and the samples with convex and concave curvatures with a large radius have the lowest roughness.

The difference in roughness between the convex and concave samples with the smallest curvature radius is outstanding. According to both forms, the roughness from cutting with one pass is higher than with two and three. This point can be an essential factor when there are convex and concave curves simultaneously in two regions of a contour form, where the radius of the curvature is small. In such a case, using an electrical discharge method, in terms of the number of passes, it is impossible to create uniformity in the recast layer surface in the two mentioned areas. Increasing the number of cutting passes only in an area with a concave curvature may be necessary.

Due to the difference in the characteristics of the recast layer surface, different behavior can be seen for the phenomenon of wear. Regardless of the size of the radius of curvature, the distribution of roughness, in the case of concave curves, is less than the distribution of roughness in convex curves, as shown in Figure 16. Therefore, in the contours of the form with a convex curvature, there may be a more significant difference in roughness in different areas. Finally, the flat surface is also considered a curvature with a large radius. In that case, it is clear that the challenges in obtaining a uniform roughness on the recast layer surface increase as the curvature radius decreases. The numbers in parentheses after WEDM indicate the number of electrical discharge passes.

#### 3.2.4. Analysis of EDS Spectroscopy Results for Wire Electrical Discharge Machined Samples with Varied Geometries and Passes

An example of EDS spectroscopic method plots, measured for tool material (HSS), is shown in Figure 17. The analysis and “detector” settings are also shown in this figure, along with the numerical results of this method. The percentage of carbon and oxygen elements was considered to measure the surface’s oxidation amount. As the primary element in steel alloy, iron was subjected to spectroscopic analysis. Copper and zinc elements were also essential as the dominant elements in the wire used in the wire-cutting machine to determine the possibility of their remaining on the recast surface.

The samples with convex, concave, and flat curves are based on geometrical and dimensional characteristics, along with the two positions determined for implementing the EDS spectroscopy method. As mentioned before, the samples were placed on the table of the wire-cutting machine so that the tool’s cutting surface was facing up (wire entry area). The order of performing the EDS method was also the same. In order to determine the percentage of the elements that existed on the recast layer, the data obtained from the EDS analysis were reviewed and compared. Apart from the effect of the geometry of the form contour, in the comparisons between the output graphs of the EDS spectroscopic analysis, for all the geometric characteristics used, the same observations were obtained based on the number of cutting passes in the electrical discharge process.

Therefore, three graphs obtained from the EDS spectroscopy method were matched, and it was observed that there are differences in the percentage of selected elements on the recast layer surface. As seen in Figure 18, three different colors are used to display EDS charts. The red, green, and blue colors represent the EDS diagrams of recast layers with three, two, and one passes in the wire discharge process. Two elements, copper (Cu) and zinc (Zn), were used to investigate the effects of the cutting wire on the recast layer. The intensity of the presence of these two elements on the recast layer is observed from high to low for one, two, and three cutting passes. This means that when the electric discharge energy is lower in the second and third passes, the amount of wire erosion was such that its effects on the recast layer surface were smaller.

It can be seen that the “intensity” of the presence of different elements in the vertical axis is expressed based on an “arbitrary unit.” This analysis fixed the measurement method and its tools for all case studies without changing the scale. The intensity of the iron element as a base element in the recast layer of the sample made with one cutting pass was the lowest. The sample made through three passes had the highest intensity of presence. The oxygen element in the sample made with three passes also has the lowest presence intensity, and the sample made with one cutting pass has the highest presence intensity. This element was investigated to determine the surface’s susceptibility to the presence of oxide layers. Figure 18 shows that the difference between the intensity of zinc and copper elements is much more significant compared to the intensity of the presence of iron and oxygen elements. Figure 19 compares the intensity of the presence of the carbon element in the upper and lower regions of the samples that were created through the electrical discharge method with one, two, and three cutting passes. According to the presented diagram, the difference between the intensities of this element’s presence decreases with the increase in the number of passes. This means that the geometry’s effect of the contour on the change in the presence’s intensity of the carbon element in the number of high passes is insignificant. On the other hand, in the samples with concave curvature, the upper region has a more significant presence of carbon than the lower region. At the same time, this trend is reversed in samples with convex curvature. Compared to the intensity of the presence of carbon in HSS that was not affected by the electrical discharge process, the difference between the intensity of the presence of carbon (entirely in the upper and lower regions) is more visible in the sample with the largest curve radius. Observable levels of the intensity of the carbon element in the samples with concave curvature have more changes compared to the intensity of the mentioned element in bulk (HSS), which was tangibly different from the bulk of HSS.

Similarly, the intensity of the oxygen element was also investigated, as shown in Figure 20. As can be seen, there was generally an upward trend between the mentioned element’s intensity at the top and bottom of the sample in all samples with the projected conditions of electric discharge. It can be concluded that samples with a smaller area have a greater tendency to oxidize. The distribution in the intensity of the presence of oxygen is higher in the samples that were made through three cutting passes. From one pass to three passes, samples with concave curvature have been in a downward trend. In other words, increasing the number of passes in these contour geometries positively affects the possibility of oxidized surfaces. However, the same process has an optimum point for samples with a convex curvature, as seen in two cutting passes. In the flat sample, this trend is upward. It can be concluded that increasing the number of passes does not always positively affect the formation of oxide layers.

#### 3.2.5. Influence of Shape Contour Geometry on Recast Layer Thickness and Cutting Edge Sharpness in Wire Electrical Discharge Machining (EDM)

FESEM images of the machined surfaces revealed that the shape contour geometry affects the recast layer’s thickness. As shown in Figure 21 for convex and concave samples, the size and type of concavity or distortion of curvature in the contour of the form depends on the thickness of the recast layer. Finally, the sharpness of the cutting edge has a decisive role in the samples. Since the samples are supposed to be used as cutting tools, the sharpness of the edges is critical. Inadequate sharpness of the cutting edge leads to increased friction and subsequent premature wear, which increases machining forces and reduces the tool’s lifespan. Also, due to the fineness of the recast layer in three-pass cutting (finishing) compared to one-pass cutting (roughing), increasing the number of passes has a positive effect on reducing the thickness of the recast layer. Finally, the cutting-edge sharpness is increased by increasing the number of passes in the EDM process.

## 4. Conclusions

The basic assumption of the components for each type of profile form is based on the combination of four modes, i.e., concave curvature, convex curvature, flat surface, and oblique surface. Based on this, HSS test samples were fabricated as cutting tools with three different radii of convex curvature, three different radii of concave arc, and a flat surface. During the WEDM operation, one-pass mode was used for roughing, two passes for semi-finishing, and three passes for finishing. Furthermore, the difference between the percentage of oxygen and carbon elements on the recast layer in the two areas above the workpiece or wire entry point and the bottom area of the workpiece or wire exit point was studied. As far as surface oxidation is concerned, the lower area of the samples is more susceptible. When moving from the one-pass to three-pass manufacturing method, the samples with concave curvature show a decreasing trend in terms of susceptibility to the formation of a surface oxide layer. This means that increasing the number of passes for these contour geometries positively affects the possibility of oxidized surfaces. However, the same process has an optimum point for samples with convex curvature, which can be seen when cutting with two passes. The difference between the intensities of the presence of carbon in samples with different contour geometries is reduced by increasing the number of passes. This means that the effect of the geometry of the contour on the change in the intensity of the presence of the carbon element in the number of high passes is insignificant. On the other hand, for the samples with concave curvature, the intensity of the presence of carbon is more significant in the upper region of the samples than in the lower region. This trend is reversed for the samples with convex curvature. Finally, the direction’s effect and magnitude of the curvature and the number of passes in the electric discharge process of the wire on the recast layer are shown. It was observed that with the increase in the number of passes in WEDM, the recast layer’s thickness was reduced, and in the areas close to the site, the uniformity of the cut contour section at the wire’s entry was better than that of the areas near the wire’s exit. The magnitude of radial geometric deviations decreased from one pass to three passes. The sign of geometric deviations for convex samples was negative, and for flat, concave samples, it was positive. This implies that the radial geometric deviation reduces convexity for convex samples, and for concave samples, it increases concavity. The magnitude of deviations was observed to be more significant for convex samples than for concave ones. With an increase in curvature radius in both convex and concave cases and a closer resemblance to a flat surface, the amount of deviation in the geometric profile of the surface decreased. Although the deviation values are not the same for convex and concave curvatures, a nearly equal trend can be observed in the behavior of geometric deviation in the surface profile of the samples, which are opposites of each other. Regarding positivity and negativity, there are differences in the amount of geometric deviation in the surface profile of samples cut with a different number of passes. In the case of samples cut with one and two passes, the mentioned difference is much more significant than that of two and three passes. Therefore, the second pass significantly affected the modification of the geometric profile of the surface compared to the third pass.

## Figures and Tables

**Figure 1 micromachines-14-01976-f001:**
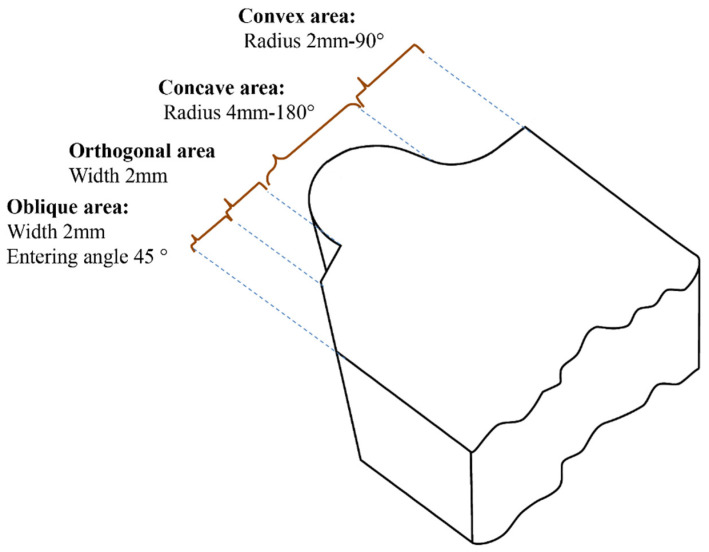
A form-cutting tool with different form profiles.

**Figure 2 micromachines-14-01976-f002:**
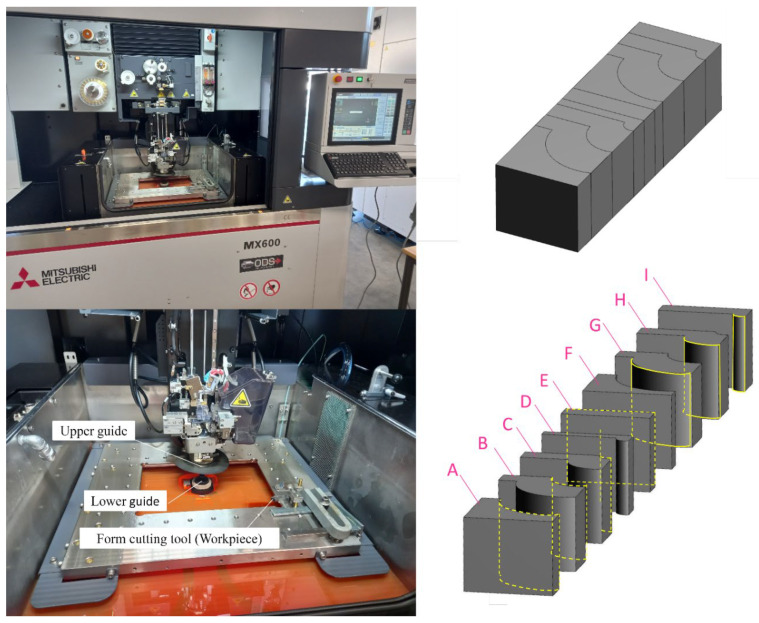
The utilized WEDM machine and produced form-cutting tools.

**Figure 3 micromachines-14-01976-f003:**
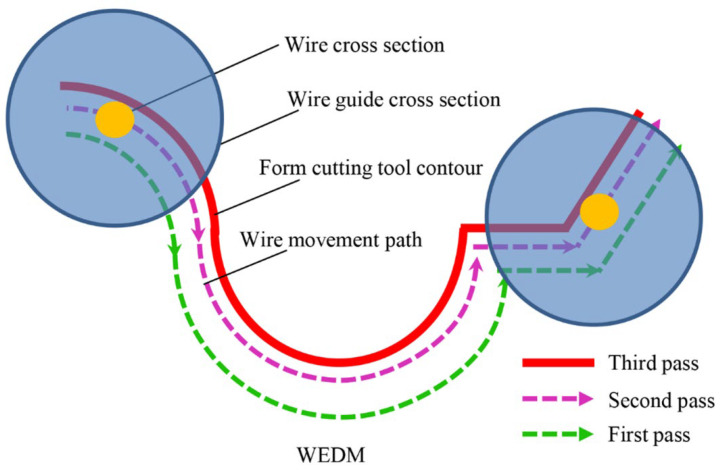
The WEDM machine path in three different strategies.

**Figure 4 micromachines-14-01976-f004:**
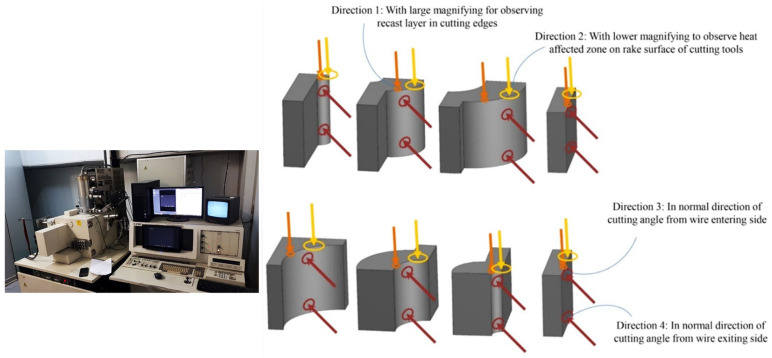
The selected directions for image scanning and the SEM instrumentation.

**Figure 5 micromachines-14-01976-f005:**
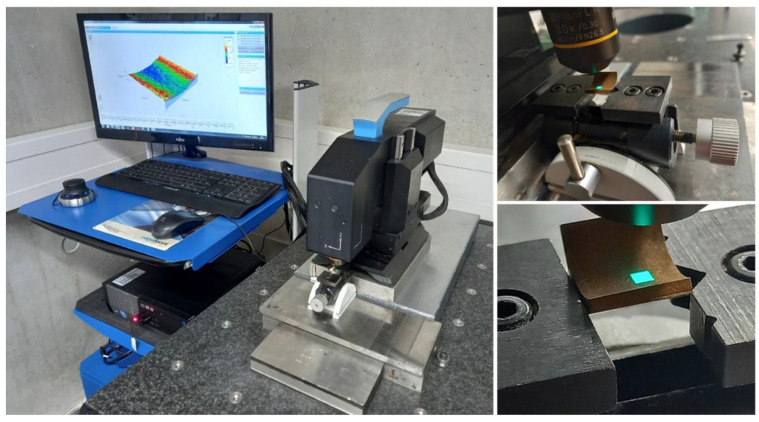
3D measurement of sample surface with confocal microscopy.

**Figure 6 micromachines-14-01976-f006:**
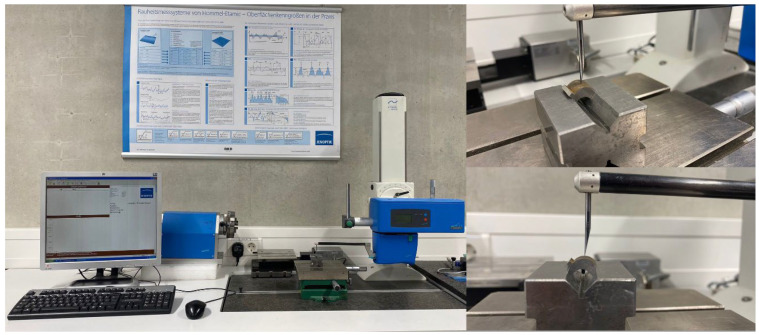
Surface profile measurements with a stylus profile meter.

**Figure 7 micromachines-14-01976-f007:**
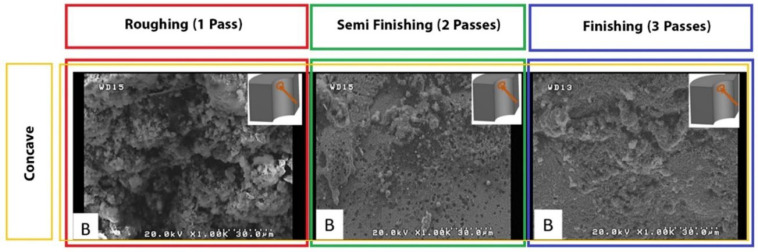
The SEM surface morphology of form-cutting tools with concave contour, induced by different WEDM steps.

**Figure 8 micromachines-14-01976-f008:**
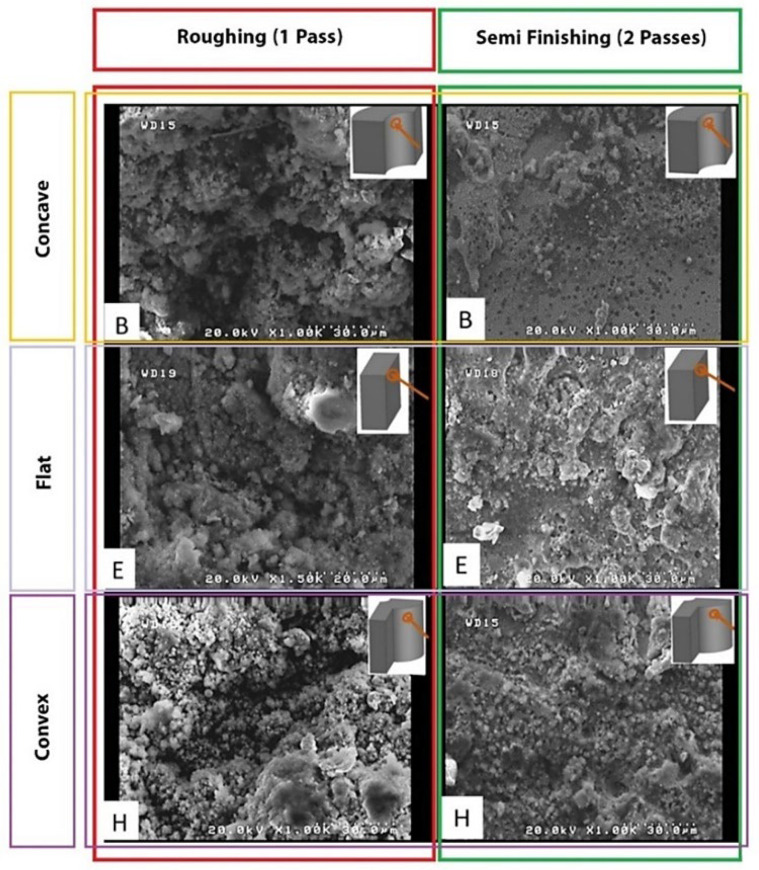
The SEM morphology surface machining of different form-cutting tools for one and two passes on the cutting surface.

**Figure 9 micromachines-14-01976-f009:**
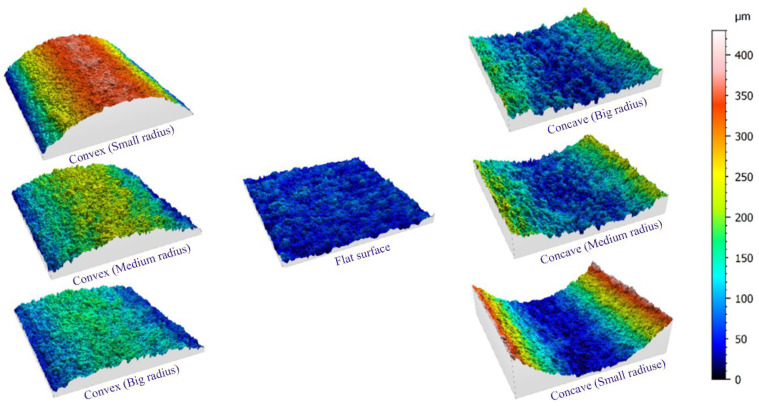
CLSM 3D surface spectrum from different geometry samples created with three-pass machining.

**Figure 10 micromachines-14-01976-f010:**
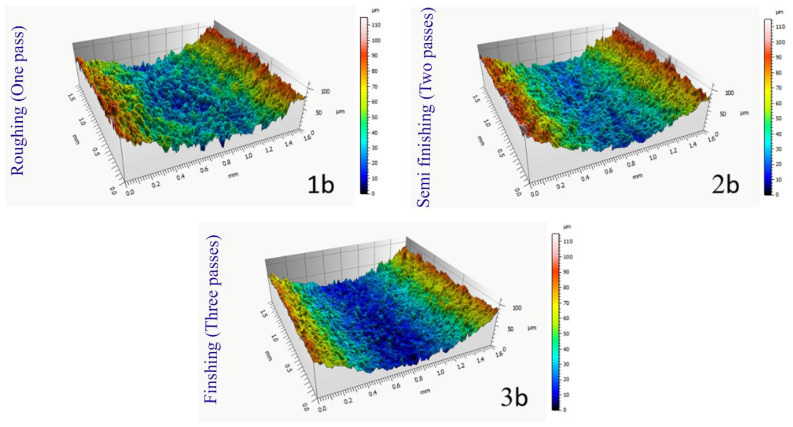
Comparison of the effect of different machining process (**1b**: single, **2b**: double, and **3b**: triple passes) on the recast layer generation represented by 3D contour scanning.

**Figure 11 micromachines-14-01976-f011:**
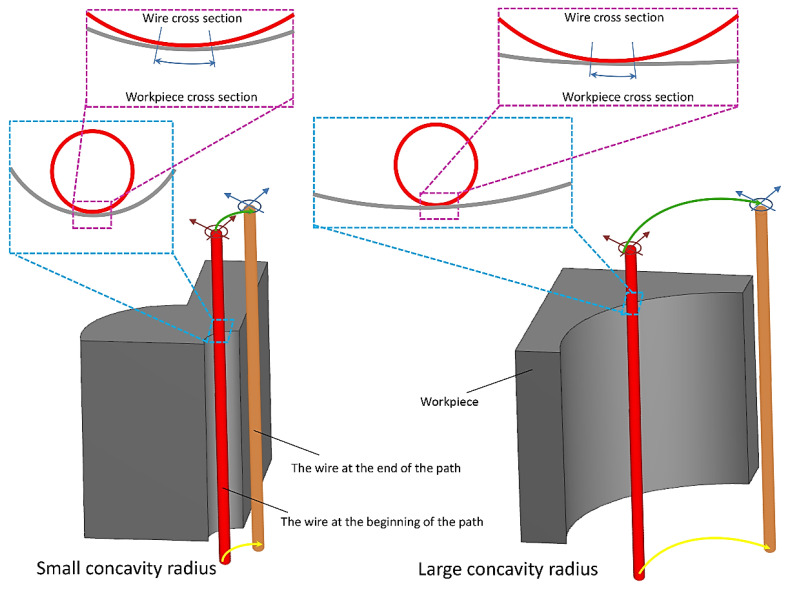
Arc length of contact between the wire and the workpiece surface with concavity radius.

**Figure 12 micromachines-14-01976-f012:**
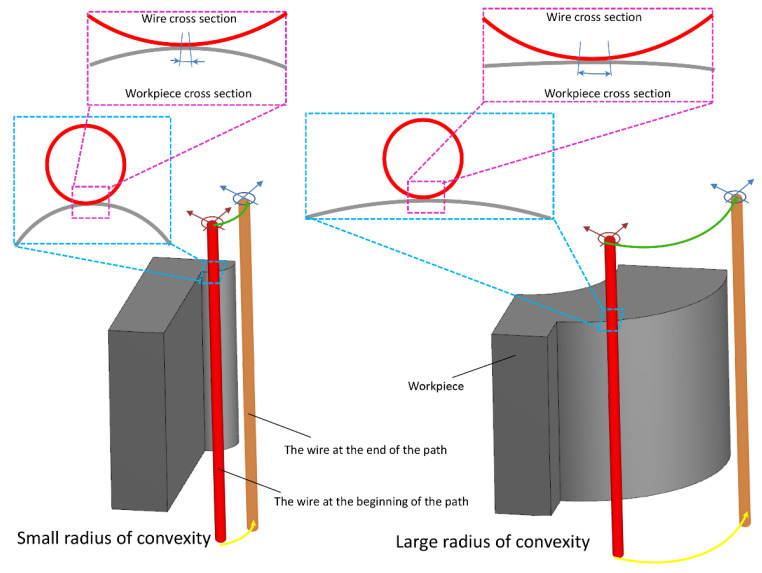
Arc length of contact between the wire and the workpiece surface with convexity radius.

**Figure 13 micromachines-14-01976-f013:**
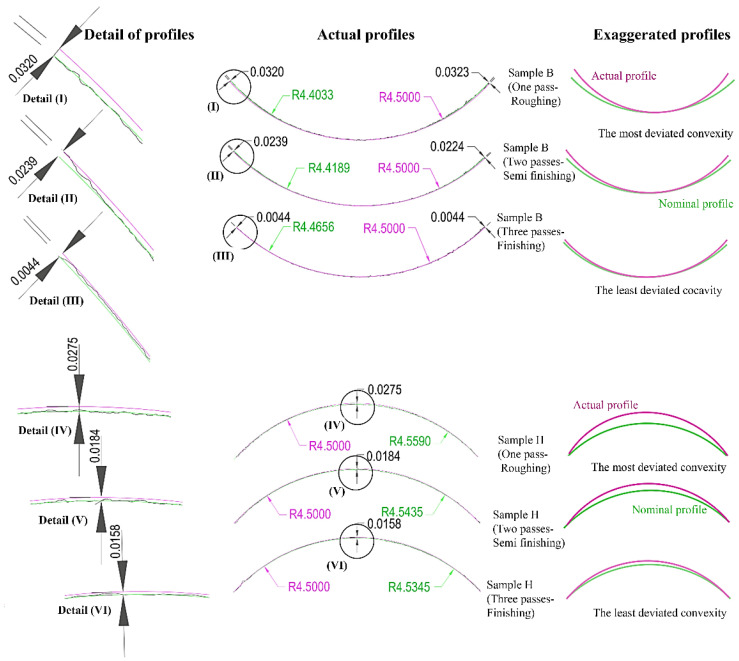
Effect of number of WEDM passes on geometric accuracy, form deviation of convex and concave curvatures.

**Figure 14 micromachines-14-01976-f014:**
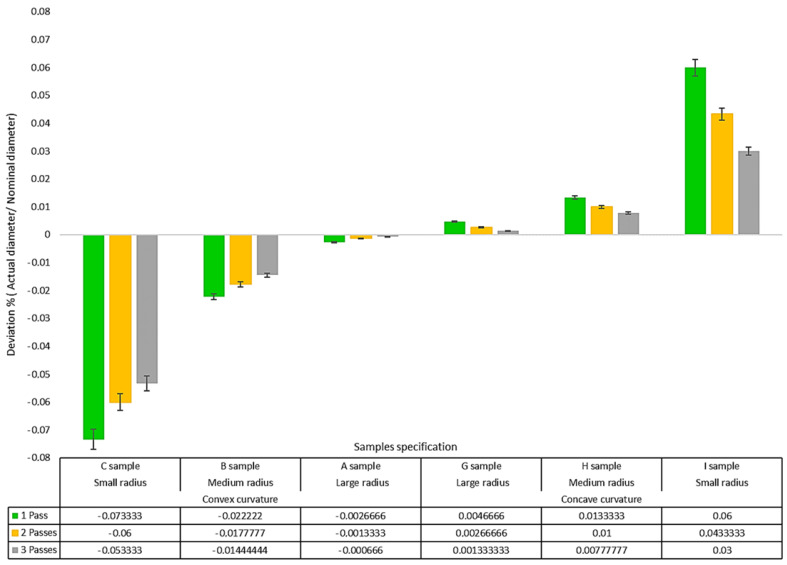
The comparison of radial geometric deviations for various samples.

**Figure 15 micromachines-14-01976-f015:**
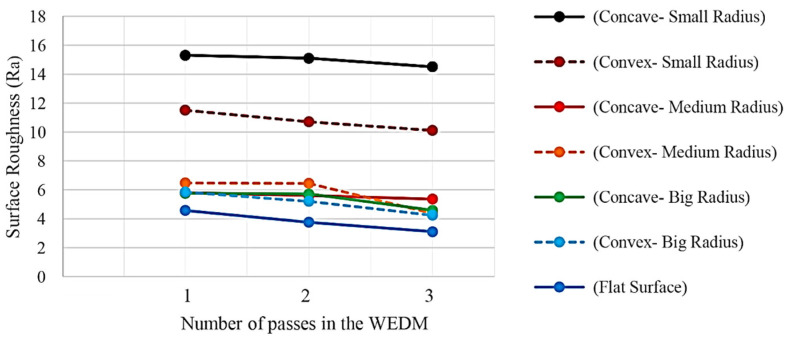
The Ra roughness criterion values for samples with different geometry profiles and three electrical discharge methods in terms of the number of cutting passes.

**Figure 16 micromachines-14-01976-f016:**
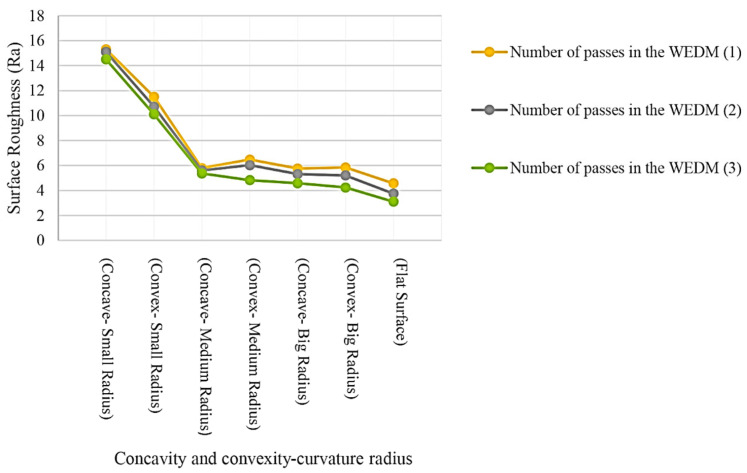
Effect of form curvature on the surface roughness based on Ra criterion for fabrication of different samples through WEDM method.

**Figure 17 micromachines-14-01976-f017:**
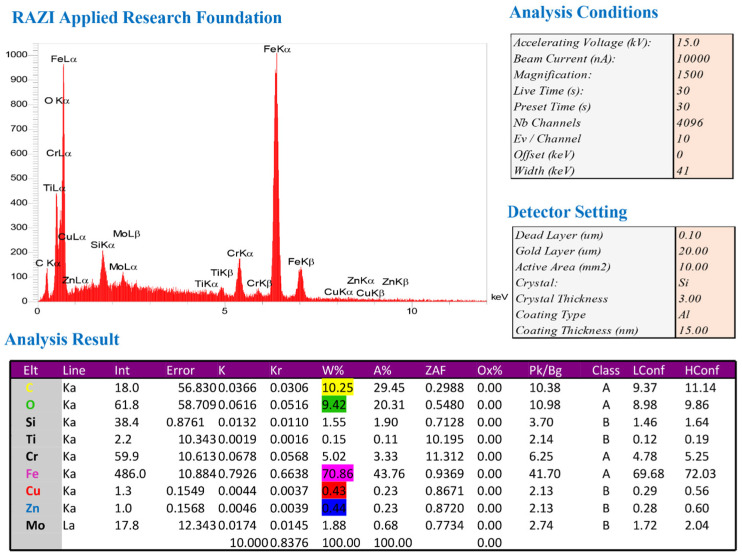
The plot of the EDS spectroscopic method. Bulk material was used for fabrication of form-cutting tools.

**Figure 18 micromachines-14-01976-f018:**
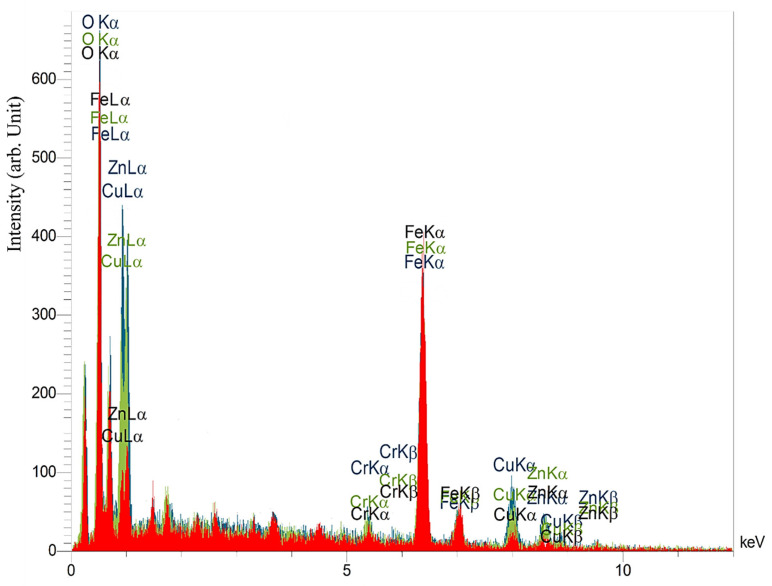
The plot of the EDS (EDAX) spectroscopic results from the same geometry samples and different machining surface finishing.

**Figure 19 micromachines-14-01976-f019:**
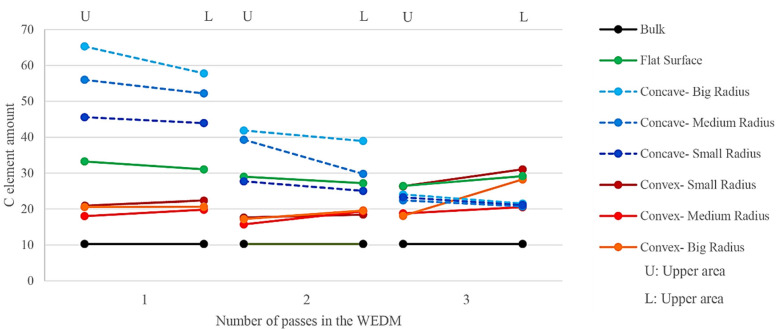
Comparison between the intensity of the presence of the carbon element in the upper and lower regions of the samples that were created through the electrical discharge method with one, two, and three cutting passes.

**Figure 20 micromachines-14-01976-f020:**
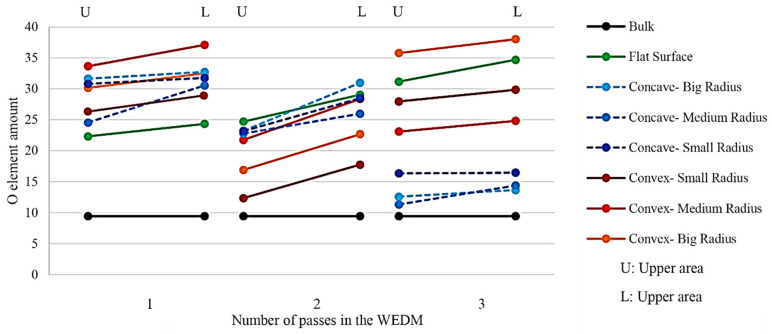
Value percentage of oxygen distribution in the upper and lower regions of the samples that were created through the electrical discharge method with one, two, and three cutting passes.

**Figure 21 micromachines-14-01976-f021:**
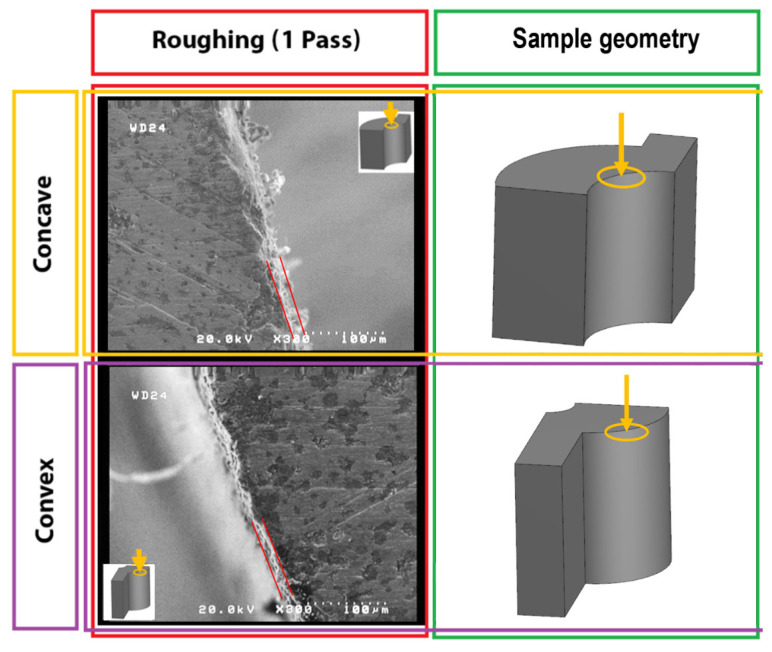
The FESEM images of the thickness of the recast layer on machined surface based on the convexity and concavity of the form contour.

**Table 1 micromachines-14-01976-t001:** The chemical composition and material properties of utilized HSS [28].

Böhler Grade	Chemical Composition in %							Standards		
	C	Cr	W	Mo	V	Co	Others	DIN/EN		AISI
Conventional high-speed steel										
Böhler S200	0.76	4.1	18	-	1.1	-	-	<1.3355>	HS18-0-1	T1

**Table 2 micromachines-14-01976-t002:** The WEDM process parameters for one-, two-, and three-pass strategies.

Tool	Brass—Wire 0.25 mm
Workpiece	HSS: VCN 150 (12 mm)
Parameter	Specification of spark energy (1 pass): Pass No. 1: IP: 14, ON time: 0.75 μs, OFF time: 30 μs, V_G_: 80 v, WT: 1 N/m, WS: 8.4 m/min, DF: 8 (L/m), FA: 10 mm/min, H: 0.156 mm
	Specification of spark energy (2 passes): Pass No. 1: IP: 14, ON time: 0.75 μs, OFF time: 30 μs, V_G_: 80 v, WT: 1 N/m, WS: 8.4 m/min, DF: 8 (L/m), FA: 10 mm/min, H: 0.186 mm Pass No. 2: IP: 9, ON time: 0.75 μs, OFF time: 30 μs, V_G_: 20 v, WT: 1.4 N/m, WS: 9 m/min, DF: 0 (L/m), FA: 2.775 mm/min, H: 0.140 mm
	Specification of spark energy (3 passes): Pass No. 1: IP: 14, ON time: 0.75 μs, OFF time: 30 μs, V_G_: 80 v, WT: 1 N/m, WS: 8.4 m/min, DF: 8 (L/m), FA: 10 mm/min, H: 0.186 mm Pass No. 2: IP: 9, ON time: 0.75 μs, OFF time: 30 μs, V_G_: 20 v, WT: 1.4 N/m, WS: 9 m/min, DF: 0 (L/m), FA: 2.775 mm/min, H: 0.140 mm Pass No. 3: IP: 4, ON time: 0.45 μs, OFF time: 30 μs, V_G_: 14 v, WT: 1.4 N/m, WS: 6.8 m/min, DF: 0 (L/m), FA: 2 mm/min, H: 0.128 mm
Dielectric	Oil—oil bath

IP: Intensity of power (base of the Mitsubishi table), ON time: Pulse time during discharging, OFF: Pulse interval time, VG: Gap voltage during discharging, WT: Wire tension, WS: Wire traveling speed, DF: Dielectric fluid flow rate (L/m), FA: Actual feed rate of electrode in cutting direction, H: Distance between electrode center to workpiece surface.

## Data Availability

No datasets were generated or analyzed during the current study.

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
