# Peer review of "Experimental Study of the Surface Quality of Form-Cutting Tools Manufactured via Wire Electrical Discharge Machining Using Different Process Parameters"

_micromachines, 2023, doi:10.3390/mi14111976_

Round 1

Reviewer 1 Report

The content of the article is substantial, the logic is clear, and it has considerable research significance. This article has publication value and can give other researchers some useful inspirations. But I still have some questions and suggestions, and it would be great if the authors can give them consideration.

1.    The abstract mainly emphasizes the research content, but the background introduction is too simple and does not clarify the significance of the research.

2.    It is recommended to adjust the font size and format of the table in Figure 14 and the colors in Figures 19 and 20 to facilitate identification.

3.    The word ' Showed ' in the first paragraph of page 18 is incorrectly capitalized.

4.    Some of the pictures in the article are unnecessarily large, resulting in a long length and large white space in some pages. It is recommended to adjust the format.

5.    There are many contents in section 3.2, so it is suggested to optimize the content and structure of this section.

It is suggested to polish the English expression of the article. Please pay attention to grammar and diction.

Author Response

Dear Reviewer,

Greetings and respect. I appreciate the time you dedicated to reviewing our submitted manuscript and the valuable comments provided to enhance its quality. The list of changes made to the paper aligns with your insightful comments, including:

Yes

Can be improved

Must be improved

Not applicable

Does the introduction provide sufficient background and include all relevant references?

(x)

( )

( )

( )

Are all the cited references relevant to the research?

(x)

( )

( )

( )

Is the research design appropriate?

( )

(x)

( )

( )

Are the methods adequately described?

( )

(x)

( )

( )

Are the results clearly presented?

(x)

( )

( )

( )

Are the conclusions supported by the results?

( )

(x)

( )

( )

  1.   The abstract mainly emphasizes the research content, but the background introduction is too simple and does not clarify the significance of the research.

I appreciate your consideration of the thoughtful points that contribute to presenting a higher-quality and more efficient paper. Based on your insights, certain sections of the introduction have been rewritten, incorporating additional sentences to clarify the scope of each part, ensuring that the presented content is purposeful and comprehensive. This refinement aims to provide the reader with a coherent mental framework to follow.

  1.    It is recommended to adjust the font size and format of the table in Figure 14 and the colors in Figures 19 and 20 to facilitate identification.

Thank you for your careful consideration. The mentioned issue has been addressed.

    1.   The word ' Showed ' in the first paragraph of page 18 is incorrectly The . . . The mentioned issue has been addressed. Thank you for your careful consideration.
  1.    Some of the pictures in the article are unnecessarily large, resulting in a long length and large white space in some pages. It is recommended to adjust the format.

Yes, exactly, and your observation is correct. Unfortunately, even after the review, the paper had not been formatted according to the standard journal format and was sent to me in the original format. Nevertheless, considering the standard journal font, all the images have been adjusted with an appropriate size:

  1.    There are many contents in section 3.2, so it is suggested to optimize the content and structure of this section.

The section 3.2, has been reviewed and rewritten, and it has been divided into relevant sections. We appreciate the feedback tool provided by you. Changes have been made.

 Thank you for your constructive feedback, which has undoubtedly contributed to the improvement of our work.

Sincerely,

Reviewer 2 Report

Yes, I agree with the authors of the manuscript that form cutting tools are an economical choice for turning parts with defined profiles in mass production. However, this is a complex task, since there are a large number of machining methods, including additional treatment of their surface. The application of a suitable method of manufacturing shaped tools brings significant savings in machining time and at the same time achieves a higher quality of the machined surface of the tool. Therefore, from this point of view, I consider the proposal of experimental research to be very relevant. Although it is questionable whether enough experiments have been performed to draw relevant conclusions. Nevertheless, I consider the information obtained from the results of experimental research to be very valuable.

Although the manuscript is well written, some corrections need to be made and the following questions answered:

1. It is generally known that the size of the remelted layer of the machined surface after WEDM depends on the combination of the setting of the main technological parameters, many of which have a priority position. However, the value of the remelted layer of the machined surface after WEDM also depends on the mutual combination of the properties of the tool electrode and the process medium (dielectric). Therefore, it is advisable to indicate parameters related to the dielectric, such as temperature, flushing pressure, etc., in the manuscript.

2. Table 2 seems confusing.

3. The surface roughness Ra in the range of 2.5 to 15, I assume that μm (because the parameter is missing in tab. 15) is abnormally high for WEDM. What is the reason for the abnormally high roughness of the eroded surface in your experiment?

4. Why were these three types of surfaces chosen for the experiment - concave and convex arcs, and flat? Why, for example, was the internal and external sharp edge not chosen, on which the recast layer will show much more than with a rounded surface?

5. There is no discussion at the end of the manuscript, whether the results of the experimental findings are applicable to the machining of other materials, in the case of other types of wire electrodes.

6. At the same time, the conclusion of the work lacks a discussion in which the authors evaluate the conclusions in the context of the existing knowledge. At the conclusion of the thesis, it is very important to qualitatively or quantitatively emphasize the points of agreement or disagreement between the results in this thesis and the cited references in the manuscript.

The contribution is processed at a good level and after modifications and additions it can be published in the journal Micromachines.

Author Response

Dear Reviewer,

Greetings and respect. I appreciate the time you dedicated to reviewing our submitted manuscript and the valuable comments provided to enhance its quality. The list of changes made to the paper aligns with your insightful comments, including:

  Yes Can be improved Must be improved Not applicable
Does the introduction provide sufficient background and include all relevant references? ( ) (x) ( ) ( )
Are all the cited references relevant to the research? ( ) (x) ( ) ( )
Is the research design appropriate? ( ) (x) ( ) ( )
Are the methods adequately described? ( ) ( ) (x) ( )
Are the results clearly presented? ( ) (x) ( ) ( )
Are the conclusions supported by the results? ( ) ( ) (x) ( )
  1. It is generally known that the size of the remelted layer of the machined surface after WEDM depends on the combination of the setting of the main technological parameters, many of which have a priority position. However, the value of the remelted layer of the machined surface after WEDM also depends on the mutual combination of the properties of the tool electrode and the process medium (dielectric). Therefore, it is advisable to indicate parameters related to the dielectric, such as temperature, flushing pressure, etc., in the manuscript.

With gratitude for highlighting the insightful points that assist us in presenting a higher-quality and more efficient paper, I would like to inform you that, according to your suggestion, in Table 2, the parameter related to the dielectric fluid flow rate (DF: Dielectric fluid flow rate (L/m)) has been provided. Regarding the temperature of the dielectric fluid, considering the use of the immersion method, there was not a significant difference in the temperatures of the available sections compared to the usual temperature of the dielectric fluid. To precisely determine the working temperature in the spark zone, an inquiry was made to the machine manufacturer, Mitsubishi (with the presentation of all cutting parameters) but they refused to provide information. . Additionally, measurement tools such as thermocouples and methods based on thermal cameras were not utilized in this research.

2. Table 2 seems confusing.

Thank you for your feedback, and I apologize for the abundance of information due to the various cutting parameters mentioned in this table. Nonetheless, respecting your opinion, I have separated the sections related to the cutting process with one, two, and three passes

3. The surface roughness Ra in the range of 2.5 to 15, I assume that μm (because the parameter is missing in tab. 15) is abnormally high for WEDM. What is the reason for the abnormally high roughness of the eroded surface in your experiment?

Appreciate your insightful question. The reason for not employing adaptive control feed conditions or the specialized Mitsubishi option called Feed Varey in this study is precisely because the authors did not intend to use these features. The wire cut machine used in this research, especially being a relatively modern model from Mitsubishi, benefits from advancements in generator design and electronic components. It has the capability to make instantaneous adjustments to the machine's tuning parameters for cutting a specific material (E-PACK: Electro package).

The first mentioned option dynamically controls and optimizes the feed rate instantaneously through adaptive control. The second option, in case of difficulty in cutting and having a low feed rate, swiftly transforms the E-PACK from the actual material of the workpiece to similar materials with almost identical cutting conditions. In the current project, since the primary goal was to compare the actual surfaces obtained with different geometries and passes, none of these options were utilized. Essentially, the perspective presented in this paper regarding wire cutting did not focus on minimizing surface roughness; rather, it centered on examining the impact of workpiece geometry and the number of passes on the recast layer

4. Why were these three types of surfaces chosen for the experiment - concave and convex arcs, and flat? Why, for example, was the internal and external sharp edge not chosen, on which the recast layer will show much more than with a rounded surface?

Thank you for your intelligent and professional perspective. It's necessary to explain that the current article is just a part of a larger project on forming cutting tools and shaping operations, which has explored various dimensions over approximately four years of effort. In this project, various dimensions of analysis have been pursued, including the formation of chips in forming tools, analysis of fractures and the direction of chip twists, permanent and non-permanent chip separations, the impact of cutting conditions on chip formation, the influence of cutting conditions on wear and tool life of forming cutting tools, the effect of geometric shapes on machining forces, and much more. These analyses, both mathematical, geometric and kinematicaly, have been validated through practical experiments.

In fact, one of the significant aspects emphasized in this research work has been the acute angles and corners (edges), both external and internal, which are crucial in cutting tools. However, due to the constraints of content and the limitations of the current article's length, it was not possible to include this important topic. Ultimately, the authors decided to address this critical aspect in a separate article. Please find an academically suitable English translation of the provided text.

5. There is no discussion at the end of the manuscript, whether the results of the experimental findings are applicable to the machining of other materials, in the case of other types of wire electrodes.

The question is very pivotal. Considering the approach of the current article, which aims to develop cutting tools, the initial investigation was conducted to identify materials, aside from HSS, that are produced as tools through the WEDM process. It was observed that for materials such as SS, HSS, WC, PCD, and recently ECCVD (Electroconductive Chemical Vapor Deposition) Diamond, which is currently under research by us, due to acceptable Material Removal Rate (MRR) and the necessity of addressing economic concerns, the use of wire electrodes continues to be prevalent in the WEDM process. Therefore, the basis for material selection is established based on the use of conventional wire electrodes

6. At the same time, the conclusion of the work lacks a discussion in which the authors evaluate the conclusions in the context of the existing knowledge. At the conclusion of the thesis, it is very important to qualitatively or quantitatively emphasize the points of agreement or disagreement between the results in this thesis and the cited references in the manuscript.

Since in all conducted research, the role of input parameters in the electrical discharge method on the surface properties of the recast layer was discussed, and conversely, in this study, the input parameters were consistent, with only changes in the flat geometry, concave or convex curvature geometry of the parts, and apart from the geometry, their size was considered as a variable. There is no possibility to confirm or reject the findings of past research. In fact, most articles have pointed out that with an increase or decrease in a parameter or a set of parameters, what result is produced on a specific material in the recast layer. Only one article had focused on curvature and corners, in which again, the size or geometry of the workpiece was not a variable, but the correction of the geometric accuracy of the arc through the optimization of conventional wire-cut parameters was the goal. Therefore, the authors decided to present the results of this article independently.

Thank you for your constructive feedback, which has undoubtedly contributed to the improvement of our work.

Sincerely,

Reviewer 3 Report

1.     Put statistical findings of your main achievement in the work in the abstract.

2.     Add a clear research gap or problem statement in the introduction section.

3.     Add all papers published in the last ten years on tool wear and include types of wear encountered in machining operation.

4.     Can the authors provide table 2 in a more user-friendly way for readers.

5.     The Methods and Materials section more elaborate to make it easy for the readers.

6.     The insert formation information is not enough in materials.

7.     No need for the SEM machine image in Figure 4, Better to insert information there.

8.     Would be better to put some images of the cutting insert at each stage of fabrication or at the final stage.

9.     How you will compare your results with the available commercial insert used in machining processes?

10.  Can authors justify their claim of oxidation of the insert surface based on published literature?

11.  The conclusion is not enough and needs more information.

12.  The form tools need to be tested for turning operation and see it performance in machining. Also, the author can compare it with the available tools.

Author Response

Dear Reviewer,

Greetings and respect. I appreciate the time you dedicated to reviewing our submitted manuscript and the valuable comments provided to enhance its quality. The list of changes made to the paper aligns with your insightful comments, including:

  Yes Can be improved Must be improved Not applicable
Does the introduction provide sufficient background and include all relevant references? ( ) ( ) (x) ( )
Are all the cited references relevant to the research? ( ) ( ) (x) ( )
Is the research design appropriate? ( ) ( ) (x) ( )
Are the methods adequately described? ( ) ( ) (x) ( )
Are the results clearly presented? ( ) ( ) (x) ( )
Are the conclusions supported by the results? ( ) ( ) (x) ( )

Comments and Suggestions for Authors

  1. Put statistical findings of your main achievement in the work in the abstract.

Thank you for your insightful and valuable feedback, which has assisted us in the pursuit of presenting a higher-quality and more informative article for our esteemed readers. It is important to inform the respected reviewer that in the current research work, statistical methods were not employed. Instead, by encompassing all possible variations of input parameters, experiments were conducted in a full factorial design. Additionally, the following aspects were measured and examined:

    - Surface roughness of the samples
    - Geometric deviations of the samples concerning their ideal geometry
    - Percentage of alloying elements present in the recast layer
    - Percentage of the existence of surface oxidation layer in the recast layer

Furthermore, experiments were performed at three levels concerning the number of cutting passes:

    - WEDM one pass
   -  WEDM two passes
   -  WEDM three passes

Finally, experiments were conducted for seven different geometries and sizes of workpiece samples:

    - Convex curvature (small radius)
    - Convex curvature (medium radius)
    - Convex curvature (large radius)
    - Flat surface
    - Concave curvature (large radius)
    - Concave curvature (medium radius)
    - Concave curvature (small radius)

Due to the multitude of experiments and measured parameters, it was not feasible to present numerical or percentage results in the abstract. Therefore, the authors refrained from including them.

  1. Add a clear research gap or problem statement in the introduction section.

With gratitude for your constructive feedback, the mentioned issue has been addressed as the final paragraph in the introduction

"As a research gap or problem statement, it needs to be stated that the influence of the WEDM process parameters on the surface quality of parts made of HSS and their form accuracies, considering various contours, has not been studied yet. The present study is focused on reducing the adverse effects of the recast layer induced by WEDM on the form cutting tools made of HSS. The basic component types of profile forms in form cutting tools are summarized by the combination of four modes, i.e., concave and convex arcs, and flat and oblique surfaces. Form cutting tools made of HSS were fabricated with three different radii of convex curvature, three different radii of concave curvature, and a flat surface. During the WEDM operation, one pass mode was used as roughing, two passes as semi-finishing, and three passes as finishing. Furthermore, the surface quality of the recast layer at two areas, namely the wire entry point and the wire exit point, was studied. Finally, the effect of the direction, size of the curvature, and the number of passes in the wire electric discharge process on the recast layer on the surface of the form cutting tools have been analyzed."

  1. Add all papers published in the last ten years on tool wear and include types of wear encountered in machining operation.

With respect to your opinion, the authors find it necessary to clarify that the current paper does not delve into the various phenomena associated with post-machining effects on cutting tools and their cutting edges. Rather, it focuses on a stage prior to machining, specifically the tool manufacturing phase. At this point, no machining parameters have been determined, and no tools have been employed for machining, allowing us to address the issue of wear. In essence, this paper is a part of a larger research project concerning cutting tools used in form machining, which covers various aspects, including post-machining conditions. However, due to the extensive content, the authors have chosen to solely present the achievements related to tool manufacturing in this paper

  1. Can the authors provide table 2 in a more user-friendly way for readers.

Thank you for your feedback, and I apologize for the abundance of information due to the various cutting parameters mentioned in this table. Nonetheless, respecting your opinion, I have separated the sections related to the cutting process with one, two, and three passes

  1. The Methods and Materials section more elaborate to make it easy for the readers.

Yes. Thank you .The Methods and Materials section has been reviewed and rewritten, and it has been divided into relevant sections. We appreciate the feedback tool provided by you. Changes have been made.

  1. The insert formation information is not enough in materials.

I appreciate the time you've dedicated to reviewing this article. As explicitly detailed in the Material and Methods section, this research exclusively utilizes High-Speed Steel (HSS) bulk material, and no inserts are employed. All relevant information pertaining to HSS is comprehensively outlined in Table 1. 

  1. No need for the SEM machine image in Figure 4, Better to insert information there.

Thank you. The sole purpose of Figure 4 was not merely to present a microscopic image. The crucial aspect of this figure lies in illustrating the chosen directions for SEM imaging of the recast layer. Given the presence of various arcs with different sizes, the normals of each surface were used as the principal direction for studying the recast layer, while the tangents served as the direction for investigating the thickness of the recast layer.

  1. Would be better to put some images of the cutting insert at each stage of fabrication or at the final stage.

Yes, you are absolutely right. In fact, our intention was also to provide suitable images of the HSS machining stage for the fabrication of form cutting tools in our articles. Unfortunately, due to the type of Wire EDM machine used for this task being an Oil bath type, and complete submersion dielectrik being employed, it was not possible to capture images. The presence of the orange-colored oil, as seen in Figure 2, serving as the dielectric fluid, prevented such documentation.

  1. How you will compare your results with the available commercial insert used in machining processes?

Thank you for your question. The authors find it necessary to elaborate on two aspects in response to this question. Firstly, as paper has explained and is evident in the article text, no inserts have been used in this research. The second point is that due to the presence of various geometries resulting from the combination of flat, sloped, concave, and convex conditions, each form insert can be unique. Depending on production volume, these form inserts may be manufactured using production methods ranging from single-piece to batch or mass production. Therefore, the WEDM method is utilized. For conventional commercial inserts simillar ISO or standard inserts, processes like powder metallurgy and some followed by grinding process will be employed. In these cases, there is no recast layer, and the geometry of the insert does not undergo significant changes that would affect the manufacturing method and results (outputs) significantly.

  1. Can authors justify their claim of oxidation of the insert surface based on published literature?

Thank you for your question. If your reference is to a proof and witness indicating the formation of surface oxidation on the recast layer, it's worth mentioning that there are numerous articles discussing this phenomenon. The authors considered it self-evident and did not use a specific article to justify its existence, as the presence of surface oxidation after WEDM cutting has been extensively discussed in various publications. However, if your inquiry pertains to the impact of different parameters on the level of surface oxidation, it should be noted that in this paper, the influence of none of the WEDM input parameters on the outputs at the cutting surface, including the recast layer, was discussed. Instead, the focus was on the geometric aspects of the workpiece. Regarding this, we invite your attention to a paper directly addressing the impact of WEDM process parameters on surface oxidation:

K Mouralova, T Prokes and L Benes ,"Analysis of the oxide occurrence
on WEDM surfaces in relation to subsequent surface treatments", 
J Mechanical Engineering Science, 0(0) 1–13, IMechE 2019
Article reuse guidelines: sagepub.com/journals-permissions
DOI: 10.1177/0954406219884974
journals.sagepub.com/home/pic

  1. The conclusion is not enough and needs more information.

In line with your suggestion, the aspects related to geometric deviations have been added to the conclusion section and are visible in the article:

"The magnitude of radial geometric deviations decreases from one pass to three passes. The sign of geometric deviations for convex samples is negative, and for flat concave samples, it is positive. This implies that for convex samples, the radial geometric deviation reduces convexity, and for concave samples, it increases concavity. The magnitude of deviations was observed to be greater for convex samples than for concave ones. With an increase in curvature radius in both convex and concave cases, and a closer resemblance to a flat surface, the amount of deviation in the geometric profile of the surface decreased. Although the deviation values are not the same for convex and concave curvatures, a nearly equal trend can be observed in the behavior of geometric deviation in the surface profile of the samples, which are opposites of each other. In terms of positivity and negativity, there are differences in the amount of geometric deviation in the surface profile of samples cut with different passes. In the case of samples cut with one and two passes, the mentioned difference is much greater than that of two and three passes. Therefore, the second pass had a significant effect on the modification of the geometric profile of the surface compared to the third pass."

Thank you for your constructive feedback, which has undoubtedly contributed to the improvement of our work.

Sincerely,

Round 2

Reviewer 3 Report

Accepted in the current form and need proofreading before final submission.